

# Evaluating the Utility of Active Microwave Observations as a Snow Mission Concept Using Observing System Simulation Experiments

Eunsang Cho[1,2], Carrie M. Vuyovich[1], Sujay V. Kumar[1], Melissa L. Wrzesien[1,2], Rhae Sung Kim[1,3]

[1]Hydrological Sciences Laboratory, NASA Goddard Space Flight Center, Greenbelt, MD, USA
[2]Earth System Science Interdisciplinary Center, University of Maryland, College Park, MD, USA
[3]Goddard Earth Sciences Technology and Research II, University of Maryland, Baltimore County, Baltimore, MD, USA

*Correspondence to*: Eunsang Cho (eunsang.cho@nasa.gov)

**Abstract.**

As a future satellite mission concept, active microwave sensors have the potential to measure snow water equivalent (SWE) with advantages including finer spatial resolution and improved capabilities in deeper snowpack and forest-covered

areas as compared to existing missions (e.g., passive microwave sensors). In mountainous regions, however, the potential utility of spaceborne active microwave sensors for SWE retrievals particularly under deep snow and forest cover has not been evaluated yet. In this study, we develop an observing system simulation experiment (OSSE) that includes the characterization of expected error levels of the active microwave-based volume-scattering SWE retrievals and realistic orbital configurations over a western Colorado domain. We found that active microwave sensors can improve a root mean

square error (RMSE) of SWE by about 20% in the mountainous environment if the active microwave signals with a mature retrieval algorithm can estimate SWE up to 600 mm of deep SWE and up to 40% of tree cover fraction (TCF). Results also demonstrated that the potential SWE retrievals have larger improvements in tundra (43%) snow class, followed by boreal forest (22%) and montane forest (17%). Even though active microwave sensors are known to be limited by liquid water in the snowpack, they still reduced errors by up to 6-16% of domain-average SWE in the melting period, suggesting that the

SWE retrievals can add value to meltwater estimations and hydrological applications. Overall, this work provides a quantitative benchmark of the utility of a potential snow mission concept in a mountainous domain, helping prioritize future algorithm development and field validation activities.

## 1 Introduction

Global distribution of seasonal snow is a critical component of the Earth's water and energy cycles (Barnett et al., 2005;

Pulliainen et al., 2020; Sturm et al., 2017). Seasonal snow covers up to 50 million $km^2$ of the Northern Hemisphere in winter, and about 17% of the world's population relies on meltwater from seasonal snow that replenishes reservoir storage and





groundwater for natural and human systems (Bormann et al. 2018; Li et al. 2017; Immerzeel et al., 2020). However, spatially distributed information on snow water equivalent (SWE; the amount of water stored in the snowpack) across the globe is limited, particularly in complex terrain such as mountainous regions where a large portion of the snowpack is commonly

distributed. In general, mountains and remote regions lack in-situ SWE networks across the globe (Dozier et al. 2016). Even if there are relatively dense ground measurement networks, the in-situ observations have limited spatial representativeness (e.g., automated snow pillow stations in the Snow Telemetry network represent ~ 3 m by 3 m area approximately), providing limited information on the spatial distribution of SWE, particularly in heterogeneous terrain (Molotch and Bales, 2005).

Historically, a series of satellite-based passive microwave radiometers have been used to develop spatially distributed snow

depth and SWE information, such as the Special Sensor Microwave Imager (SSM/I) and SSM Imager/Sounder (SSMIS) aboard the Defense Meteorological Satellite Program (DMSP) series of satellites and the National Aeronautics and Space Administration (NASA) Advanced Microwave Scanning Radiometer for Earth Observing System (AMSR-E) onboard the Aqua satellite and the Japan Aerospace Exploration Agency (JAXA) AMSR2 onboard the Global Change Observation Mission 1st-Water (GCOM-W1) (Cho et al., 2017; Derksen et al., 2005; Foster et al., 2005; Vuyovich et al., 2014). However, the

passive microwave satellite-based SWE retrievals have a coarse spatial resolution (~ 25 km) and large SWE uncertainties in various snow and land conditions, which often limits their utility for water supply assessments and operational weather prediction applications (Lettenmaier et al. 2015; Carroll et al., 1999). The passive microwave retrieval algorithms do not perform well under a deep snowpack approximately greater than approximately 200 mm SWE (the so-called "saturation effect") because the microwave radiation at higher frequency does not decrease with increasing SWE (Derksen et al., 2010;

Dong et al., 2005). Errors in SWE retrievals generally increase with increasing forest density (Cho et al., 2020; Foster et al., 2005; Vander Jagt et al., 2013). Passive microwave radiation is also highly sensitive to small amounts of liquid water content in the snowpack (Kang et al., 2013; Walker &Goodison, 1993), hampering accurate SWE retrievals under wet snow conditions. Though data assimilation efforts such as the GlobSnow project (Pulliainen et al., 2020; Takala et al., 2011), have attempted to integrate passive microwave brightness temperature measurements and in-situ observations to generate improved SWE,

coverage over mountainous regions is still lacking in these products due to large uncertainties over these areas (Larue et al., 2017; Pulliainen et al., 2020). Therefore, global coverage of SWE information is still elusive despite the long legacy of passive microwave instruments.

Active microwave sensors (e.g. synthetic aperture radar; SAR) have a great potential to measure SWE with advantages including higher spatial resolution and improved capabilities in deeper snowpack and forest cover (Lievens et al., 2019; Rott

et al., 2010; Tsang et al., 2022). SWE retrievals using X- and/or Ku-band radar is a viable approach as a global satellite mission concept because these measurements are sensitive to SWE through the volume scattering properties of dry snow. In recent decades, the potential for radar to retrieve SWE has been explored in the snow remote sensing community. The Cold Regions Hydrology High-resolution Observatory (CoReH2O) mission concept, a dual-mode high-frequency (X- [9.6 GHz] and Ku-band [17.2 GHz]) SAR, was proposed to the European Space Agency (ESA) in response to the 2005 Earth Explorer Core



Mission Call. This mission was selected by ESA for feasibility studies (Phase A) in 2009 but was not selected for further implementation (Rott et al., 2010). In addition, as part of the NASA Snow and Cold Land Processes (SCLP) Mission (National Research Council, 2007; Yueh et al., 2009) and the Cold Land Processes Experiment (CLPX) activities (Cline et al., 2009; Elder et al., 2009; Tedesco et al., 2005), a high-frequency SAR and high-frequency (K- and Ka-band) passive microwave radiometer were explored. Recently, Environment and Climate Change Canada (ECCC) in partnership with the Canadian

Space Agency (CSA) initiated a new dual Ku-band frequency (13.5 and 17.2 GHz) SAR mission (Terrestrial Snow Mass Mission; TSMM) concept study (Derksen et al., 2019).

The series of radar mission development activities in recent decades, which include multi-year field and airborne campaigns, has played a major role in the considerable progress achieved towards the use of radar remote sensing techniques not only to estimate snow microstructure and SWE but also to identify retrieval uncertainties in diverse regions such as deep

snow and forests (King et al., 2018; Nagler et al., 2008; Rott et al., 2010; Rutter et al., 2019; Zhu et al., 2018). Historically, the detectable SWE threshold of radar techniques is considered to depend on snow stratigraphic properties but is approximately 300 mm with a frequency range from X to Ku band (Nagler et al., 2008; Rott et al., 2010), though the threshold was determined based on limited observations. This is primarily due to the saturation of the volume scattering. As a future direction of the algorithm development at X- and/or Ku-band, Tsang et al. (2022) mentioned that the co-polarization X-band backscatter signal

could be used for estimating deeper SWE (> 300 mm) along with a multilayer algorithm (King et al., 2018; Rutter et al., 2019). As a different frequency approach, Lievens et al. (2019) show the capability of C-band cross-polarization backscatters (5.4 GHz) from Sentinel-1 for measuring deep snow depths (e.g. more than 2 m in Figure 7 of Lievens et al., 2019). For dry snow, the empirical change detection algorithm can retrieve snow depth up to 5 m deep at less than 1 km spatial resolution over mountain ranges (Lievens et al., 2022).

For forest effects, Nagler et al. (2008) found that the presence of dormant herbaceous vegetation has a small influence on the backscattering of the active microwave signals but does not affect the sensitivity to SWE. However, the backscatter signal may be affected in coniferous forests based on simulation studies. In the case of low fractional cover (< 25%) in coniferous forests, simulations with a radiative transfer model found that the snow backscatter dominated the radar signal (Macelloni et al., 2001; Magagi et al., 2002). When the forest fraction increases, the sensitivity of the backscatter to SWE generally decreases.

Tsang et al. (2022) demonstrated that at Ku-band frequency (17.2 GHz; wavelength: 1.74 cm), the Ku-band wave can travel in straight lines as rays through the gaps in trees. The Ku-band waves could pass through the gaps like LiDAR (light wave detection and ranging) which is considered to be able to penetrate forest canopies. This suggests that the SWE retrievals in areas with up to 40% of TCFs could be achievable with efforts to account for the three-dimensional structure of the canopy for a more detailed and accurate assessment of the impact of forest type and density on the SWE sensitivity. Considering that

forested regions are a significant portion of the global snow-covered extent (Rutter et al., 2009; Kim et al., 2021), even slight advancements in retrieval algorithms for improved handling of forest effects will directly help extend valid coverage of the SWE measurements as a global snow mission. However, the utility of active microwave SWE measurements with the degrees





of retrieval's limitations is not well quantified in prior studies. A formal assessment of the utility of hypothetical active microwave sensors for SWE estimation under different observing conditions (e.g., deep snow, dense forests, and the presence
of liquid water) is, therefore, needed to establish the potential benefits of such future sensors and to set priorities related algorithm developments.

An Observing System Simulation Experiment (OSSE; Arnold and Dey, 1986; Masutani et al., 2010) is a modeling and data assimilation-based approach that is often used to assess the utility of spaceborne observations from proposed designs of new satellite missions before the instruments are deployed. OSSEs enable the quantification of the utility of spaceborne
observations and help in the design and configuration of future missions (Crow et al. 2005; De Lannoy et al. 2010; Garnaud et al. 2019; Kumar et al., 2014; Kwon et al., 2021; Nearing et al. 2012). Specifically for SWE, De Lannoy et al. (2010) used an OSSE to explore techniques for downscaling coarse-scale SWE products to the underlying fine-scale model state variables within a data assimilation system. More recently, Kwon et al. (2021) conducted light detection and ranging (LiDAR) OSSE to quantify the accuracy requirement of spaceborne LiDAR snow depth retrieval which provides its beneficial impact on SWE
and hydrologic variables within a land surface model. They found that synthetic LiDAR observations provided utility in assimilation processes when the realistic snow depth retrieval's error standard deviation is lower than 60 cm. Like the current study, Garnaud et al. (2019) used an OSSE to estimate the potential value of the Ku-band radar mission concept for the Environment and Climate Change Canada- Canadian Space Agency (ECCC-CSA) Terrestrial Snow Mass Mission (TSMM). They used an OSSE to inform on the optimal mission configuration (i.e., resolution, revisit time, and snow mass retrieval
uncertainty) using a testbed in southern Quebec, Canada. In the non-mountainous, forested domain, this study found that bias in a baseline SWE simulation was largely reduced by improving the revisit frequency (e.g., 93% with 1-day revisit time), and systematic errors were also reduced by a higher revisit frequency as well as an increased resolution (1 km rather than 2 or 10 km spatial resolution).

The main objective of this study is to quantify the usefulness of volume-scattering SAR SWE retrievals for improving
spatially distributed characterization of snow conditions through an OSSE setup over a mountainous region of western Colorado. Specifically, we focus on the SWE retrieval utility over deep snow and forest-covered regions. We introduce the study area and describe our OSSE design, including the main steps in sections 2 and 3, respectively. The results of OSSE performances are reported in section 4. Lastly, we discuss implications and limitations and provide concluding remarks in section 5.

## 2 Study domain: western Colorado

The western Colorado region is selected as the OSSE domain providing a representative continental mountainous region (**Figure 1**). The study area includes four seasonal snow classes: tundra (7.1%), boreal forest (14.3%), montane forest (44.9%), and prairie (28.9%). The seasonal snow classification is based on the 1-km new seasonal snow classification developed by

Sturm and Liston (2021). The elevation over the domain ranges from 1400 m to 4000 m (41% of the domain area is between
1400 - 2500 m, 33% between 2500 - 3000 m, and 26% between 3000 – 4000 m) which is based on a 1-km elevation map
derived from the 'Native' United States Geological Survey's (USGS) Shuttle Radar Topography Mission (SRTM) elevation
data (Farr et al., 2007). Tree cover fraction (TCF; %) ranges from 0 to 80 % (49% of the domain area with 10% or lower, 13%
with 10 – 20%, 19% with 20 – 40%, 14% with 40 - 60%, and 4.3% with 60% and higher TCFs). The upscaled 1-km TCF map
is derived from the 30-m resolution global tree cover data developed by the University of Maryland (Hansen et al., 2013) using
a bilinear resampling approach. This domain also includes previous field campaign experiment locations such as NASA-
NOAA Cold Land Processes Field Experiment (CLPX; 2001 - 2003) and NASA SnowEx field campaign (2017, 2020, and
2021).

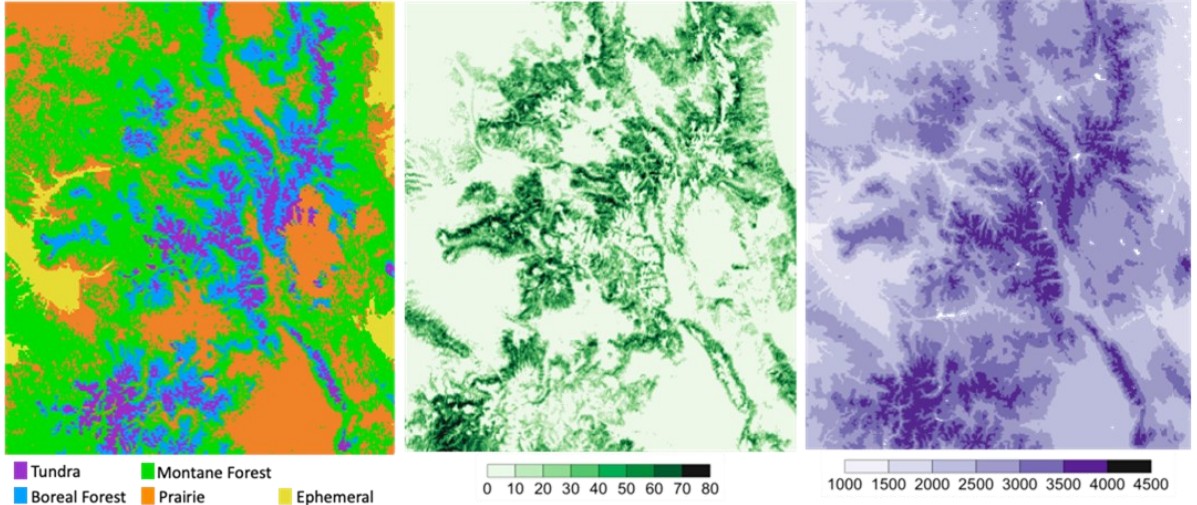

**Figure 1**. (a) Sturm and Liston's seasonal snow classification, (b) tree cover fraction (%) map from University of Maryland,
and (c) elevation (m) of the study area over the western Colorado

## 3 Observing System Simulation Experiment design

An OSSE is a data assimilation-based modeling approach that is typically used to quantify the utility of satellite
observations from proposed instrumental designs of a new mission before the instrument is deployed. The OSSE performed
in this study focuses on quantifying the beneficial impacts of hypothetical X- and/or Ku-band active microwave SWE
observations with different levels of retrieval uncertainties at a 1 km spatial resolution. The OSSE setup includes three main
elements: 1) the Nature Run (NR), 2) Open Loop (OL), and 3) Data Assimilation (DA) simulations with synthetic observations
(**Figure 2**). The NR is the calibrated land surface model (LSM) simulation which is considered the "truth" in the OSSE
framework (section 3.2). The OL is an uncalibrated LSM simulation as the default configuration (section 3.3). The DA
scenarios are simulation results assimilating hypothetical synthetic observations with different error constraints with OL.
Detailed information about synthetic observations and DA are provided in sections 3.4 and 3.5, respectively. To develop





realistic synthetic observations, we apply a subsampling method to obtain a realistic active satellite viewing area from
hypothetical satellite-based radar using the Trade-space Analysis Tool for Constellations (TAT-C) simulator (Le Moigne et
al., 2017). The study period used in the analysis is the winter season from 1 October 2016 to 31 May 2017 which experienced
moderate snow conditions and provided sufficient differences between NR and OL SWE making the OSSE setup effective for
quantifying improvement. A model time step of 15 min was used and daily averaged model outputs were saved for analysis.
For all experiments, relevant physical parameterization options of the Noah-MP version 4.0.1 were used as listed in **Table 1**.
Then we develop 24 DA experiments from synthetic observations with assumptions of uncertainty related to deep snow and
forest coverage. Detailed descriptions of how to apply those limitations to DA experiments are given below.

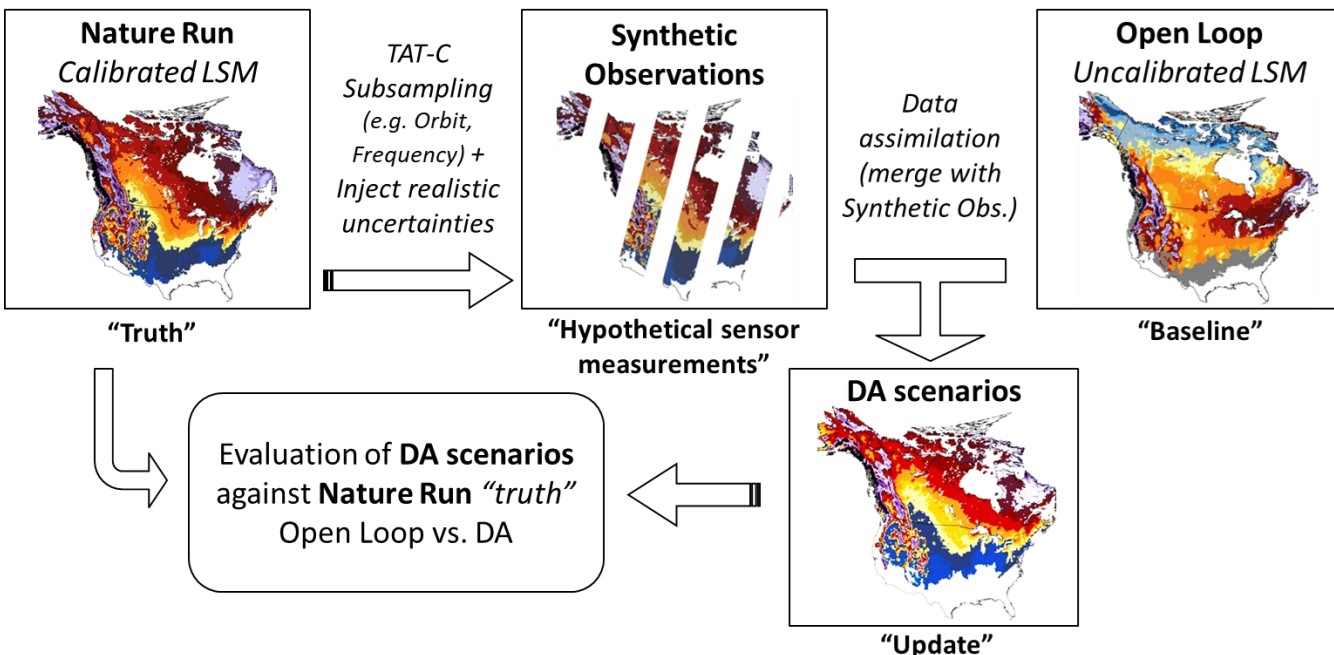

**Figure 2.** Schematic diagram of the synthetic, observing system simulation experiment (OSSE) setup of this study. A Nature Run (NR)
indicates synthetic truth simulation, and open-loop (OL) and data assimilation (DA) are model simulations without and with assimilation
of the synthetic snow water equivalent (SWE) retrievals, respectively, derived from the NR.

**Table 1**. Relevant physical parameterization schemes of Noah-MP (version 4.0.1) used in the Observing System Simulation Experiments
(OSSEs)

| Physical process | Option used | References |
|---|---|---|
| Lower boundary condition of soil temperature | Original Noah scheme | - |
| Supercooled liquid water (or ice fraction) in frozen soil | NY06 | Niu and Yang (2006) |
| Frozen soil permeability | NY06 | Niu and Yang (2006) |
| Ground snow surface albedo | Biosphere-Atmosphere Transfer Scheme | Yang and Dickinson (1996) |
| Precipitation partitioning into rainfall and snowfall | Jordan91 | Jordan (1991) |



| Snow and soil temperature time scheme | Semi-implicit | - |

### 3.1 NASA Land Information System and Noah-MP land surface model

The OSSE simulations were conducted using the NASA land information system (LIS; Kumar et al. 2006; Peters-Lidard
et al. 2007), which is a software framework for high-performance land surface modeling and data assimilation experiments.
Within LIS, we employed the Noah multi-parameterization (Noah-MP) LSM version 4.0.1 (Niu et al., 2011; Yang et al., 2011).
Noah-MP was developed based on the original Noah LSM (Ek et al., 2003) with augmented representations of biophysical
and hydrological processes. Noah-MP includes a multilayer snowpack representation (up to 3 layers) to simulate the physical
processes of varying snow density over time, allowing patchy snow cover to evolve as a function of snow depth and vegetation
type. The model simulates snowpack liquid water retention, refreezing of meltwater, and frost/sublimation, all of which are
important for the accurate characterization of snow conditions. The model also accounts for snow age, grain size growth, and
the effect of impurities on snow evolution. Previous studies found that Noah-MP has superior performance to the original
Noah LSM and other LSMs for simulating SWE (Cho et al., 2022; Kim et al., 2021; Minder et al., 2016).

### 3.2 Nature run (Synthetic Truth)

We used the calibrated Noah-MP simulation at 0.01° spatial resolution (~ 1 km) described in Wrzesien et al. (2022) for the
NR in this experiment. The meteorological forcing data for the simulation was the North American Land Data Assimilation
System phase 2 (NLDAS-2; Xia et al., 2012). In that study, the optimization and uncertainty subsystem (Kumar et al., 2012)
within LIS was used to calibrate Noah-MP SWE against estimates from the observation-based University of Arizona gridded
snow dataset (UA; Zeng et al., 2018). For the optimization, Wrzesien et al. (2022) used a genetic algorithm to calibrate 23
model parameters related to snow parameterizations that are hard-coded into the default Noah-MP configuration, and an
additional snowfall scaling term was included to address precipitation biases in the meteorological forcing data (Enzminger et
al., 2019; He et al., 2019; Henn et al., 2018; Raleigh et al., 2015; Schmucki et al., 2014). The calibration approach generated
spatially varying parameters, as compared to the spatially uniform values in the default Noah-MP. When evaluated against
both UA and SNODAS estimates, the calibrated simulation decreased domain-averaged temporal RMSE and bias for SWE
and snow depth, relative to the default Noah-MP configuration, for the same western Colorado domain used here in the OSSE.
Further, the snowfall scale term was shown to be important for increasing the magnitude of snow accumulation, especially in
higher-elevation grid cells.

### 3.3 Open Loop simulation

The model run without assimilation, called the OL, is conducted with meteorological boundary conditions from Modern-
Era Retrospective Analysis for Research and Applications forcing data (MERRA2, version 2) produced by NASA's Global
Modeling and Assimilation Office (Gelaro et al., 2017). MERRA2 forcing data, which have a native spatial resolution of 0.5°
latitude by 0.625° longitude (roughly 50 km), are downscaled to a 1 km grid of the model setup within LIS. Note that the OL



configuration has two primary differences relative to the NR setup: 1) the boundary conditions are different (OL: MERRA2 vs. NR: NLDAS-2) and 2) the OL uses the default configuration of Noah-MP, whereas NR uses the calibrated, spatially distributed parameters developed by Wrzesien et al. (2022).

### 3.4 Synthetic observations with TAT-C subsampling

We develop synthetic SWE observations by including factors that represent uncertainties related to snow estimation over deep snow and when vegetation is present. For deep snow, four different hypothetical limits of retrieval algorithm are considered:  200, 400, 600 mm, and no limit of SWE. The influence of forest cover is examined by considering six scenarios that limit SWE detection at different levels of forest fraction (10, 20, 40, 60, and 80 %) based on the 30-m University of Maryland Global Tree Cover Fraction (TCF) data (Hansen et al., 2013). The 24 scenarios of active microwave synthetic SWE

observations are used in the OSSE. For example, a DA run with a 20% TCF limit means that grids with >20% forest fraction are masked out from DA, assuming that the hypothetical sensor cannot measure SWE in those grids. Because active microwave sensors cannot detect SWE if the snowpack contains liquid water (Matzler, 1987; Rott et al., 2010), synthetic observations are only assimilated when the snowpack does not include liquid water content (LWC). That is, when LWC values from the OL run are positive (> 0) at certain grids and periods, corresponding synthetic observations are not assimilated with the OL run.

Unbiased random errors with zero mean and 30 mm of standard deviation expected as an error level of the SWE retrievals from previous findings (Rott et al., 2010; Garnaud et al., 2019) are applied to the synthetic observations. To support the impact of the standard deviation on SWE evaluation, different DA scenarios with different ranges of standard deviations (10, 30, 50, and 100 mm) are compared in Supporting Information (**Figures S1** & **S2**).

       To simulate the viewing extent of hypothetical X- and/or Ku-band sensors, we use the TAT-C (Le Moigne et al., 2017),

which is a NASA-developed software system specifically designed for future Distributed Spacecraft Missions (DSM). TAT-C allows for the exploration of a range of feasible design options (e.g., single vs constellation, polar-orbiting vs geostationary, low frequency vs high-frequency overpasses) to quantify measurable gains as a function of mission configuration. In this study, the orbital configuration (e.g., Keplerian elements) of a volume-scattering SAR mission is used in the orbit and coverage module to simulate the nadir position track. Then, the realistic spatial coverage and temporal frequency are simulated by

extending the ground track to a given swath width (i.e., 250 km) in the cross-track direction. In this study, the viewing extent simulation is expressed as a daily binary map (so-called "cookie cutter") marking the surface as viewed (or not) at a 1-km spatial resolution.

### 3.5 Data assimilation

For this OSSE work, the 1-dimensional ensemble Kalman filter (EnKF) method (Reichle et al., 2002) is used to assimilate synthetic SWE observations into Noah-MP. The EnKF method includes forecast and update steps. In the forecast step, an ensemble of SWE and snow depth is propagated by Noah-MP until synthetic SWE observations become available. Each ensemble member is generated by perturbing model initial conditions, boundary conditions from a meteorological forcing, and



Noah-MP model prognostic variables based on the assumption of a Gaussian distribution. The perturbation parameters used
in this study are based on earlier DA works (**Table S1**; Kumar et al., 2014; Kwon et al., 2021). Noah-MP OL was initialized
by spinning up a simulation from 1st October 2012 to 31st September 2015. After that, a 20-member ensemble run was
additionally spun up from 1st October 2015 to 31st September 2016 to establish the initial conditions of the ensemble. The OL
and DA scenarios were simulated from 1st October 2016 to 31st May 2017.

### 3.6 Performance evaluation matrices

For evaluation, the root mean square difference, RMSE, between the DA (or OL) SWE and NR SWE over a period is
quantified as follows:

$$RMSE_{DA} = \sqrt{\frac{1}{n} \sum_{t=1}^{n}(SWE_{DA,t} - SWE_{NR,t})^2} \qquad \text{Eq. (1)}$$

$SWE_{DA}$ and $SWE_{NR}$ refer to DA (or OL) SWE and the NR SWE, and $t$ is a date. The DA RMSE improvement as compared to
baseline (OL) RMSE is calculated

$$Improvement\ (\%) = (RMSE_{DA} - RMSE_{OL})/RMSE_{OL} \cdot 100 \qquad \text{Eq. (2)}$$

## 4. Results

### 4.1 Evaluation of OSSE at a domain-averaged scale

### 4.1.1 The impact of deep SWE limits

To assess the impact of SWE retrievals on regional snowpack characterization, the DA performance is quantified using
domain-averaged SWE (**Figure 3**). This figure shows domain-averaged SWE time series from NR (synthetic truth), OL
(baseline), and multiple DA scenarios with different deep snow limits from shallow (200 mm), moderate (400 mm), deep SWE
(600 mm), and no limit. The analysis also shows SWE from model integrations stratified over different elevation ranges. Note
that here we assume no limitations due to forest coverage. For the entire domain, the peak values of the domain-averaged SWE
time series of the NR and OL are around 220 mm and 160 mm in early March, respectively. The OL simulation underestimates
SWE by 27% as compared to the NR. The underestimations are partially reduced with DA scenarios, except for the DA
integration with a 200 mm limit. The DA run with a shallow SWE limit (up to 200 mm; blue line) has little impact on the
domain-average SWE and even contributes to a degradation near the peak SWE period (February and March). However, the
DA with a moderate SWE limit (up to 400 mm; cyan line) shows improvements relative to the OL SWE. This indicates that
the retrieval algorithm with an SWE range up to 400 mm would add value to domain-averaged SWE time series in such a
mountainous region. The improvement was observed even during the ablation period. As the deep snow limits further increase
(up to 600 mm and no limit), domain-averaged SWE estimates are also improved (see the pink and green lines). The capability





to characterize deep snow has a larger impact on areas with higher elevations as those regions typically have deeper snowpacks. For mid and high-elevation ranges, the DA SWEs with 600 mm and no limits show improvements, whereas little improvements

are obtained in low-elevation ranges. This indicates that a large portion of the SWE improvements for the entire domain is contributed by the high-elevation regions. When comparing the DA time series for mid-elevation and high-elevation regions, smaller differences from the NR (black line) during the melting period are observed in the high-elevation regions, likely because melt starts later in these areas. The gaps (biases) in the SWE time series between the DA with no limit and NR may be due to the limited ability to detect wet snow and the revisit frequency. Since the random errors added to the NR are centered

on zero, the random errors may not contribute to the biases found in the domain-average approach.

**Figure 3**. Domain-average SWE comparison between NR, OL, and DA experiments with different deep snow limits (200, 400, 600 mm, and no limit) for the entire domain and subareas with three different elevation ranges.





### 4.1.2 The impact of the sensor's detection capability over forest fraction

**Figure 4.** Domain-average SWE comparison between NR, OL, and DA experiments with different levels of detection capability in areas with bare ground and tree cover fraction (TCF) limits up to 10, 20, 40, 60, and 80%


To quantify the SWE characterization based on the sensor's capabilities over forest cover, domain-averaged SWE time series from DA scenarios with simulated observations capturing SWE in areas with the bare ground (i.e. ability to detect SWE in bare-ground areas only where TCF rounds off to 0%) or TCF limits up to 10, 20, 40, 60, and 80% (i.e. ability to detect SWE even in densely forested areas up to 80% TCF) are shown in **Figure 4**. The domain-averaged NR SWE (which is the "Synthetic

Truth") has a larger SWE than the DA and OL SWE throughout the whole period, and the differences are larger in the melting period than in the accumulation period. During the accumulation period, there are similar improvements in SWE (up to 25%) among the DA scenarios with different TCF limits as compared to OL, except for TCF of 0% which was similar to OL. The SWE difference between the DA scenarios slightly increases after the large melting event in March. This tendency continues until early May when there are melting events. For areas with low and mid elevations, there are small SWE differences among





most DA scenarios with different TCFs ranging from 10 to 80 %. In areas with high elevations, there appear to be larger SWE differences between the DA scenarios in April and May than at lower elevations. The SWE improvements gradually increase with the increasing detection capability of TCFs. In other words, while the SWE retrieval capability in denser forests has a lower impact on the domain-averaged SWE performance in the low and mid-elevations of this domain where there are fewer forested areas, it has larger impacts over high elevations. For areas with high elevations, the DA SWEs with 10 – 40 % of TCFs show improvements, while no improvements are obtained in lower elevation ranges, indicating that a large portion of the SWE improvement for the entire domain is from high-elevation regions.

### 4.1.3 Different performances between accumulation and melting periods

**Figure 5** provides a comprehensive comparison of RMSEs and percent improvement calculated by the time series of domain-averaged SWE between all DA scenarios and the OL simulation relative to the NR. These percentage improvements are computed at each grid and then averaged for given areas. DA performances are different between accumulation and melting periods, where, generally, the RMSEs between DA (or OL) and NR during the accumulation period are smaller than those during the melting period. While the RMSEs range from 16 mm ("no limit" for deep SWE and an ability to detect SWE up to "80% TCF") to 28 mm (200 mm SWE limit and 80% TCF) for the accumulation period, the RMSE's range for the melting period is between 33 mm to 47 mm. Note that the baseline OL simulation itself had a large difference between the two periods (accumulation: 24 mm and melting: 44 mm). The percentages of RMSE improvements calculated using Eq. (2) show relative improvements in DA scenarios from OL for a given period. As shown in **Figure 3**, the DA scenarios with a shallow SWE limit (up to 200 mm) show little impact or degradation for domain-averaged SWE estimations as compared to OL for both periods. This implies that if a satellite mission can hypothetically provide a finer spatial resolution SWE product (1 km) than current passive microwave observations (e.g. AMSR2), but still has the current limitations (200 mm deep SWE limit and 20% of TCF), the SWE estimates may not improve the domain-averaged SWE in mountainous regions. The DA scenarios with the 400, 600 mm, and no limits clearly show improvements for both periods, and the level of the improvements varies by TCFs. For the accumulation period, the RMSE errors were reduced by around 15 % (and 23%) with capabilities for up to 400 mm (600 mm) limit and 10% or larger TCFs. For the melting period, the percent improvement is relatively small, ranging from 1% (400 mm limit and TCF 10%) to 16 % (600 mm limit and TCF 80%). For the DA scenarios without deep snow limits, the improvements range from 26 to 33% and from 12 to 26% for the accumulation and melting periods, respectively. This indicates that the ability of the active SWE retrievals to handle deep snow could help achieve better estimations of SWE during a melting period.



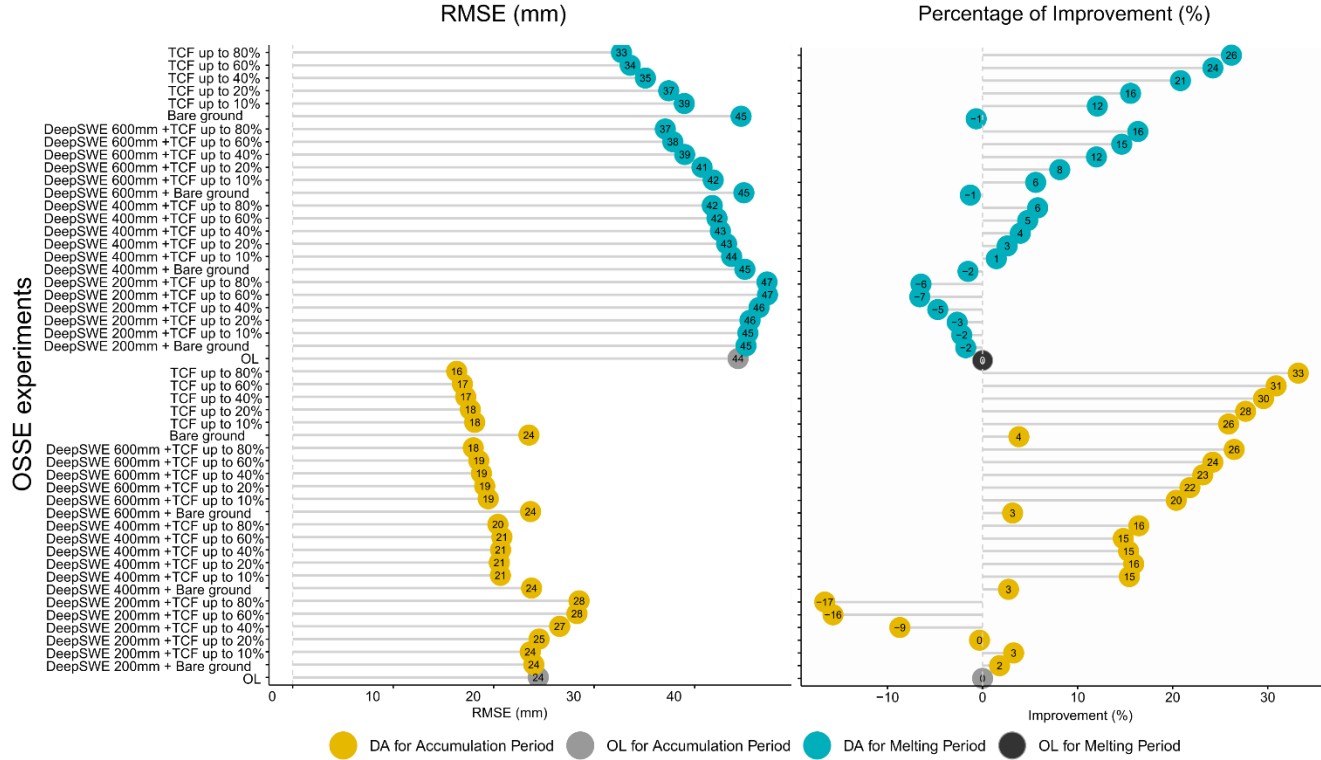

**Figure 5.** RMSEs between domain-averaged SWE estimations from the 24 DA experiments and that of the Nature run (NR) and the percentages of improvement as compared to the open loop (OL) simulation

## 4.2 Spatial evaluation of SWE performance

In this section, we evaluate the DA performance based on the spatially distributed RMSE values. **Figure 6** shows an example of spatial maps showing the annual mean SWE distributions from NR (**Figure 6a**), OL (**Figure 6b**), and a DA (**Figure 6e**) scenario, with no deep SWE limit but TCF 40%, along with a map of the number of valid days used to calculate RMSE. The annual mean NR SWE map is noticeably different from that of the OL. The annual mean DA SWE map shows similar spatial patterns with OL and NR but different magnitudes regionally. The two RMSE maps also show similar spatial patterns but of regionally different magnitudes. For this DA run, there are clear differences between the two maps over areas with TCF < 40% such as a north-central region and some southern parts of the study area (e.g. Rio Grande National Park). **Figure 7** provides a spatial comparison of RMSE between DA scenarios with the four deep SWE limits and NR. With increasing deep SWE limits, the DA's RMSEs decrease over mountainous regions where NR SWE is typically high. While there are some degradations over areas where SWEs are typically low (e.g. red color in Figure 7b), the RMSE difference maps between DA and OL demonstrate that RMSEs can be improved by more than 400 mm (areas with blue color), highlighting the importance of SWE retrievals' capability for deep snow in those mountainous environments.



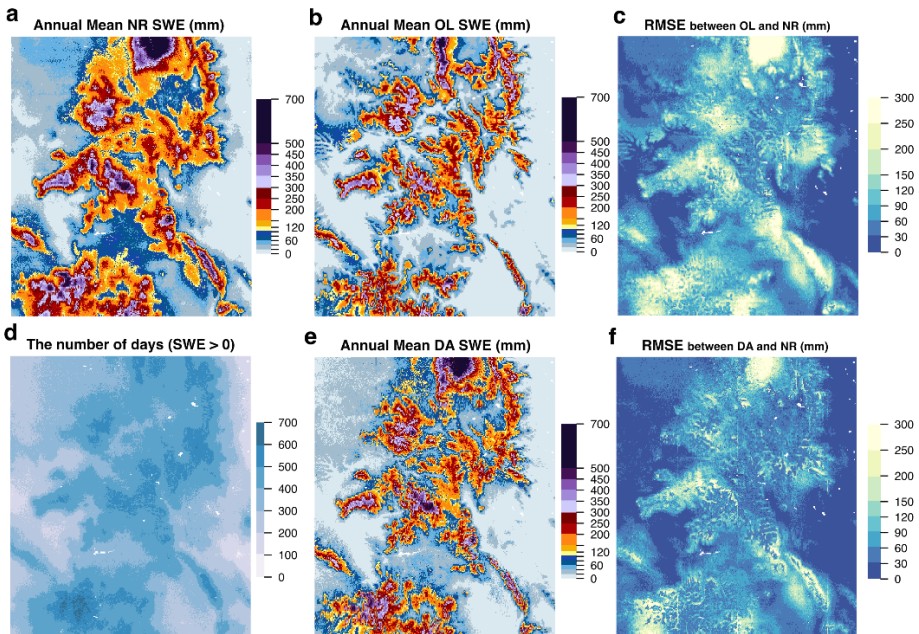

**Figure 6** The annual mean SWE maps of Nature Run (NR; left), Open Loop (OL; middle top), and data assimilation (DA) run with "no deep SWE limit but TCF limit up to 40%" (middle bottom), as an example, and the RMSE maps of OL and DA against NR

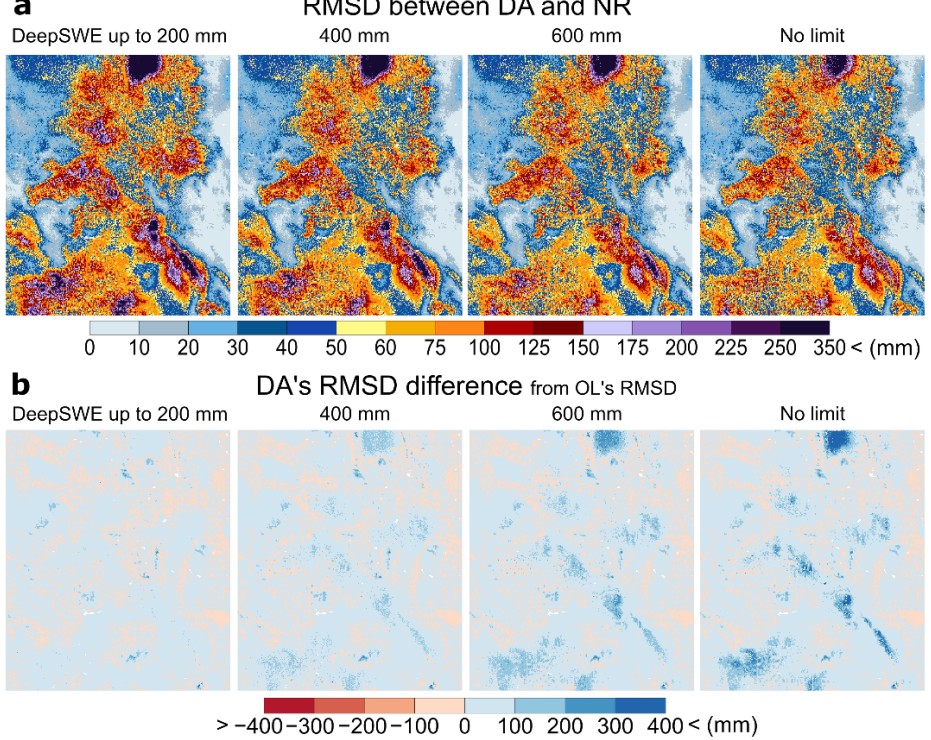





**Figure 7** (a) The RMSE maps between DA experiments with different deep snow limits (200, 400, 600 mm, and no limit) against the Nature run (NR) and (b) the four DA's RMSE difference maps from OL's RMSE. Note that the four DA scenarios are with no tree cover fraction (TCF) limit.


To quantify the improvements of 24 DA experiments relative to the OL run, RMSE comparisons between DA experiments and NR from all grid cells over the study domain are provided in **Figure 8**. The RMSE boxplot of the OL (bottom) has a range from 34 mm (lower quartile; Q1) to 112 mm (upper quartile; Q3) with a median of 67 mm (Q2). Each DA run shows different ranges of the RMSEs as compared to the OL. For example, the DA run with a 200 mm SWE limit and a TCF limit up to 20%

has slightly lower RMSEs (median: 63 mm) ranging from 28 mm (Q1) to 108 mm (Q3). For a DA run with a better capability to detect deep SWE up to 600 mm and denser TCF up to 40 %, the median RMSE decreases by 67 mm to 50 mm. If the hypothetical sensors with an ideal retrieval algorithm have a better capability to detect all deep SWE with TCF up to 80 %, the DA run has 46 mm of median RMSEs ranging from 24 mm to 76 mm, reducing by about 21 mm from the OL's RMSEs.

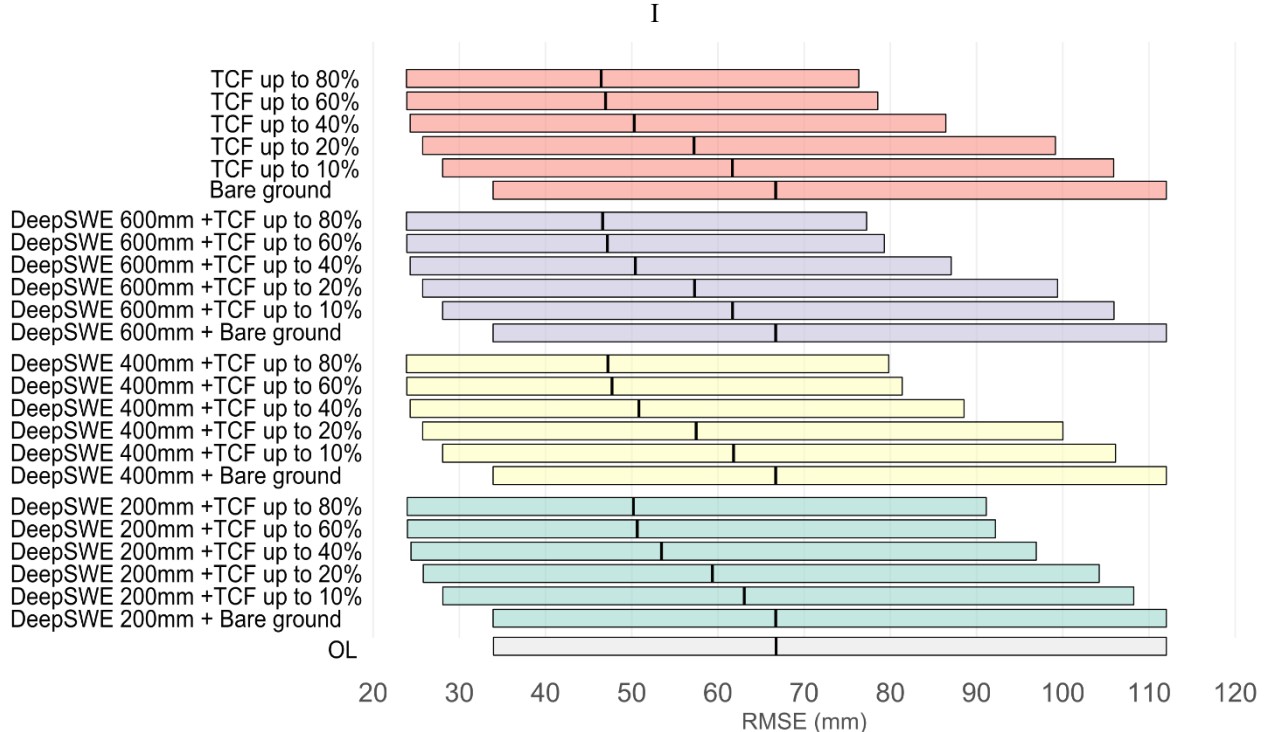


**Figure 8** Boxplots of RMSE (mm) from all grid cells between the 24 DA experiments having different combinations of deep snow and tree cover fraction (TCF) detection limits and the Nature run (NR) relative to the Open loop (OL) simulation. The black vertical lines in each boxplot indicate the median value.

To present the error improvements from each DA experiment effectively, spatial mean RMSEs and improvements (%) of RMSEs for the 24 DA experiments relative to OL RMSE are provided in **Figure 9**. The RMSEs of DAs with TCF 10% are improved by 7% (RMSE: 80 mm) to 10 % (RMSE: 73 mm) depending on the degree of deep snow limits. The DAs with TCF



80% can reduce errors by up to 25% (RMSE: 54 mm) if there is no limit with deep SWE. The DA scenarios with TCF 40% are capable of achieving up to 20% improvements in RMSE, suggesting that it would be worth improving the retrieval algorithm to detect SWE in regions with forest fractions up to 40%. To achieve around 20% of the RMSE improvements, the SWE retrievals may have to work with either 600 mm of deep SWE with TCF 40% or with 400 mm of deep SWE with TCF 60%.

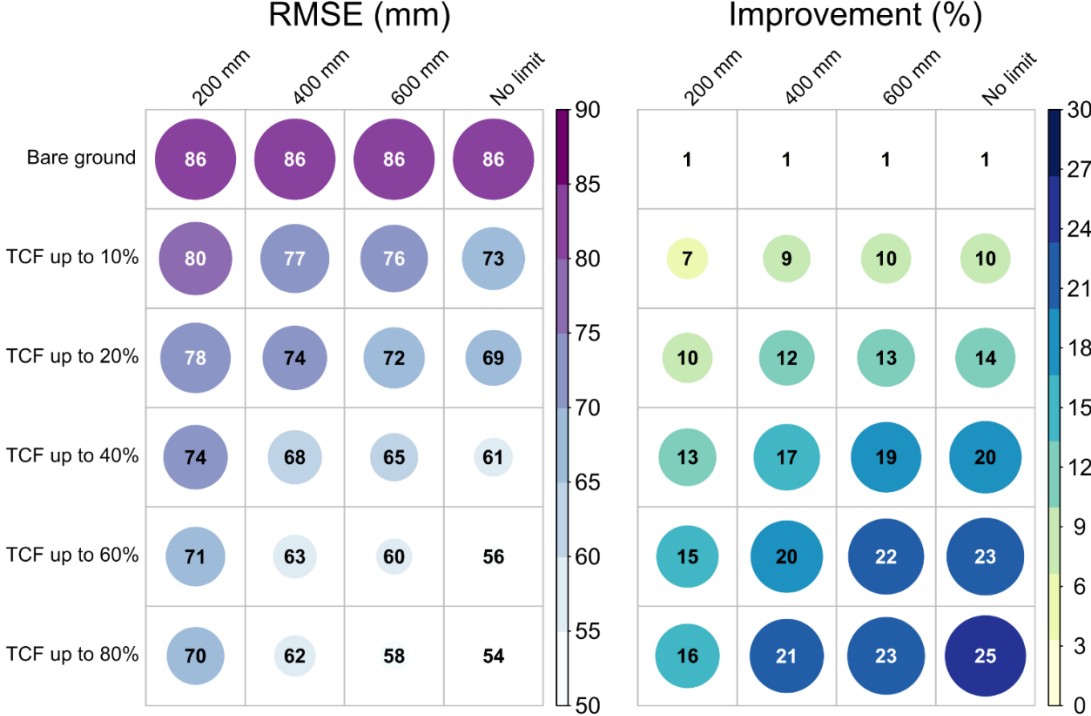

**Figure 9** Spatial mean improvement (%) of RMSEs between the 24 DA experiments with different levels of deep snow and tree cover fraction (TCF) detection limits and the Nature run (NR) relative to the Open loop (OL) simulation

### 4.3 OSSE performances by seasonal snow classes

The spatial mean percentages of the RMSE improvement by seasonal snow classification developed by Sturm and Liston (2021) are presented in **Figure 10**. The domain consists of four seasonal snow classes, tundra (7.1%), boreal forest (14.3%), montane forest (44.9%), and prairie (28.9%). To help identify spatial areas, individual maps of each snow class with different TCF ranges are included in the Supporting Information (**Figure S3**). The figure reveals that the error improvements differ by snow classification and thus different priorities for the algorithm development may be required by seasonal snow characteristics. For example, in the tundra class, there are large differences in performance between TCF 0% vs. 10%, but minimal changes are found beyond TCF 10%, due to the lack of trees in tundra environments. The ability to measure deep SWE is also important in this class because there are larger improvements with increasing deep snow thresholds, whereas there are relatively smaller improvements with different TCF levels. In boreal and montane forest classes, there are large differences



in performance between TCF 20% vs. 40%, suggesting that the capability of the SWE retrieval algorithm even up to TCF 40% can provide considerable improvement in SWE estimates in both forest environments. In the prairie class, the largest differences in performance between the deep snow limit of 200 vs. 400 mm, but minimal changes are found beyond 400 mm. This is because the prairie class typically has a shallow snowpack. Thus, a matured retrieval algorithm with active microwave

sensors detecting SWE up to 400 mm may be enough to obtain accurate SWE measurements over the prairie snow class. Overall, priorities to improve the capabilities of the retrieval algorithm for deep snow or forest areas could differ by snow class based on the mission's goal.

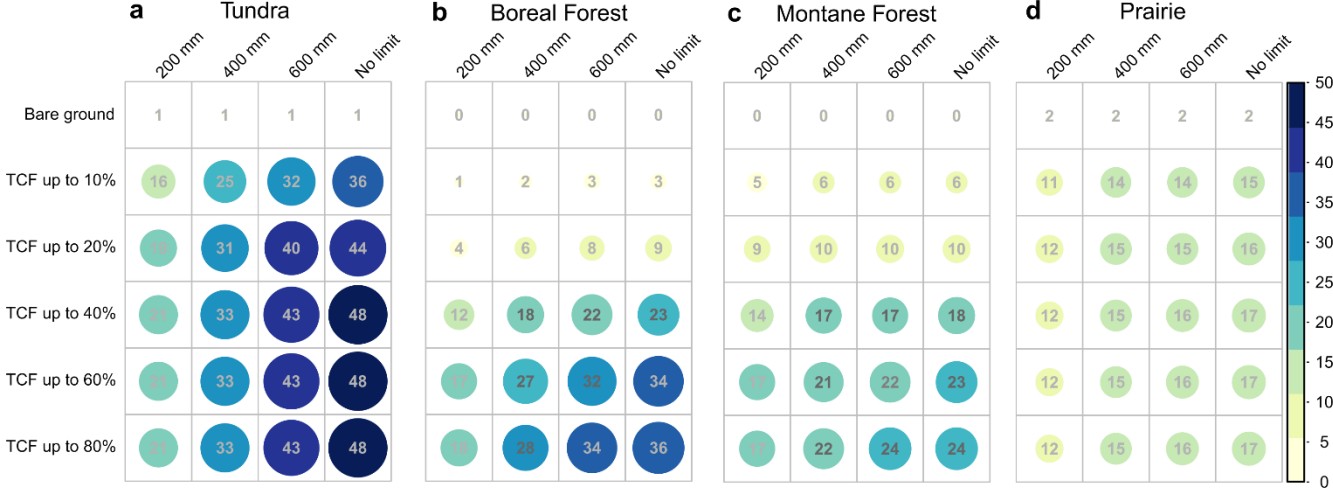

**Figure 10** Spatial mean improvement (%) of RMSEs between the 24 DA experiments with different levels of deep snow and tree cover fraction (TCF) detection limits and the Nature run (NR) relative to the Open loop (OL) simulation for four seasonal snow classes, respectively.

## 5. Discussion and Conclusion

Active microwave (radar) sensors have a great potential to measure SWE because of their sensitivity to the volume
scattering of dry snow with enhanced capabilities in deep snow and forest effects at higher resolutions (Lievens et al., 2019; Tsang et al., 2022) relative to existing missions (e.g., passive microwave sensors). The OSSE results from this study suggest that the radar snow mission may be able to reduce the RMSE by 20% in the mountainous regions if the retrieval algorithm works in snowpack environments, up to 600 mm of deep SWE with 40% of TCF. This means that the algorithm developments must focus on enhancing the retrieval skill in both deep snowpack and moderate forest fractions.

A radar-focused OSSE has been recently performed by Garnaud et al. (2019) to assess the utility of hypothetical snow observations in southern Quebec, Canada. As a part of the ECCC-CSA TSMM concept study, they conducted Ku-band radar OSSEs to quantify trade-offs between SWE performance and sensor configurations and the retrieval algorithm accuracy. There are several differences between the current study and Garnaud et al. (2019) in terms of domain characteristics, objectives, and





conclusions. While Garnaud's work focused on lower elevation (0 – 700 m elevation), forest-dominant regions with shallow
and moderate snowpack (e.g., 80 mm of the peak SWE from synthetic truth), this study focuses on a mountainous domain, in
western Colorado, with a wider range of high elevations (1000 to 4000 m) including various seasonal snow types. This domain
includes both shallow snow at lower elevations (peak SWE: 95 mm) and deep snow at high elevations (peak SWE: 430 mm
in **Figure 3**), enabling us to quantify the utility of active microwave SWE stratified over deep snow limits as well as snow
classes. The major findings from both studies complement one another. They determined the impact of different spatial
resolutions (i.e., 1, 2, vs. 10 km), revisit frequencies (i.e., 1, 3, vs. 5 days), and the retrieval algorithm accuracies. In this study,
with achievable realistic sensor configurations (1 km spatial resolution and the realistic orbital configurations for a volume-
scattering SAR mission developed using TAT-C), our study focused on the impact of potential limitations (e.g. deep snow and
forest fractions) on the SWE performance to help prioritize the algorithm developments. Our major finding is that a certain
improvement in SWE estimation in complex mountainous terrain can be achieved through improved SWE retrievals of deep
snow and snow in forested areas.

There are limitations to this study that may need to be considered in future research. First, the domain of this study (i.e.,
western Colorado) contains four seasonal snow classes and wide elevation ranges, enabling us to represent mountainous
environments and quantify approximate performances in other regions that have similar snow regimes and land surface
characteristics. However, we acknowledge that it is not enough to extrapolate our findings to global coverage of a future
mission concept. Further OSSE investigations with multiple domains in different snow climates, vegetation characteristics,
and terrain complexity (e.g., steep vs. flat terrain) will complement current efforts. Secondly, we applied a spatially constant
error across the domain. While the error (30 mm of standard deviation with zero mean) was based on the expected uncertainty
from previous studies (e.g., Rott et al., 2010), spatially and temporally dynamic error characteristics of the radar in OSSE
experiments could improve the performance assessment. At the same time, radar uncertainty in snowpack depends on the
temporal evolution of snowpack and detailed spatial features of land properties (e.g., snow microstructure, tree structures, and
canopy distribution within a grid). With ongoing efforts from current and upcoming field campaigns such as NASA SnowEx
campaigns and airborne Cryosphere-Observing SAR (CryoSAR; led by Richard Kelly at University of Waterloo), radar-snow
error characteristics will be better quantified in various environments, helping develop more realistic OSSE experiments.
Lastly, the improvement of the SWE uncertainties is inherently affected by the choice of land surface models, meteorological
boundary conditions, and spatial and temporal domains. Future studies to quantify the impact of these contributing sources on
the performance assessment will help maximize the suitability of the OSSE design.

In summary, we developed OSSEs that include characterization of expected error levels of SWE estimates and realistic
orbital configurations of anticipated sensors within NASA LIS over a western Colorado domain. We found that active
microwave X- and/or Ku-band frequencies can improve RMSE by up to 20% over western Colorado if the active microwave
signals with a mature retrieval algorithm can estimate SWE up to 600 mm of deep SWE and up to 40% of TCF. In that case,
the active microwave sensors provided larger SWE improvements in tundra (43%) and boreal forest (22%) snow classes, and
there are values in the montane forest (17%) due to deep snow capability. Active microwave sensors, known with limitations



to liquid water, can still reduce errors by up to 6-16% of domain-average SWE even in the melting period depending on TCFs, suggesting that active microwave SWE retrievals can add value for hydrological applications. Overall, this work

provides general quantification of the utility of potential radar mission concepts for SWE in a mountainous domain, helping prioritize algorithm developments and relevant upcoming field campaigns.

*Data availability*. The Global Seasonal-Snow Classification data, Version 1, is available at the National Snow and Ice Data Center (NSIDC) (http://dx.doi.org/10.5067/99FTCYYYLAQ0). The USGS 'Native' Shuttle Radar Topography Mission

(SRTM) elevation data is available at USGS Earth Resources Observation and Science (EROS) Center website https://www.usgs.gov/centers/eros/science/usgs-eros-archive-digital-elevation-shuttle-radar-topography-mission-srtm.   30-m resolution global tree cover data are available at https://glad.umd.edu/Potapov/TCC_2010/. The MERRA2 forcing dataset is available    at    the    NASA    Goddard    Global    Modeling    and    Assimilation    Office    website    (GMAO; https://gmao.gsfc.nasa.gov/reanalysis/MERRA-2/data_access/). To replicate the simulation, interested users can freely access

the NASA LIS at https://github.com/NASA-LIS/LISF.

*Author contributions*. EC conceptualized the research, led the investigation, did the formal analysis, and wrote the initial draft. CMV and SVK conceptualized the research, took responsibility for the investigation, acquired the funding and the resources, and supervised the project. RSK and MLW helped with the model simulations and investigation and provided technical and

scientific inputs. All authors reviewed and edited the paper.

*Competing interests*. On behalf of all authors, the corresponding author states that there is no conflict of interest

*Acknowledgments*. The authors acknowledge support from NASA Terrestrial NASA Hydrology (THP) Program

(NNH16ZDA001N). We are grateful to all colleagues who contributed to the SEUP project. Computing resources to run the NASA land information system (LIS) were supported by the NASA Center for Climate Simulation.

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
