# Peer review of "Evaluating the Utility of Active Microwave Observations as a Snow Mission Concept Using Observing System Simulation Experiments"

_The Cryosphere, 2022_

## Author Comment (AC1)

**Reviewer #1: Melody Sandells**

This paper demonstrates (mostly) an improvement in simulated SWE from assimilation of synthetic observations derived from an active microwave sensor through an OSSE. The analysis is based on the difference between a calibrated nature run and uncalibrated open loop run, with an assessment of the effects of SWE retrieval and forest cover fraction limits for a fixed satellite orbit and error budget configuration (with extended error budget in the supplementary material). The analysis is performed over a region in Colorado over a range of elevations, vegetation densities and snow types. The paper is very well written and easy to follow, with some excellent visual representations of the results. As the authors highlight, the results depend on the assimilation system used and this complements the Garnaud et al. (2019) study well. The results are useful to demonstrate the necessary retrieval performances for the design of retrieval algorithms and as such is worthwhile for publication.

[Answer] Thank you, Dr. Sandells, for your excellent feedback and all the valuable comments on our manuscript. We have carefully revised our manuscript based on your comments.

It would be great to elaborate on these discussion points as part of this paper:

'We found that active microwave sensors can improve a root mean square error (RMSE) of SWE by about 20% in the mountainous environment if the active microwave signals with a mature retrieval algorithm can estimate SWE up to 600 mm of deep SWE and up to 40% of tree cover fraction (TCF).' What would be really good for this paper is a small discussion of to what extent the retrieval algorithms can actually fulfil the different criteria e.g. in Figure 10, it might be possible to colour-code the boxes behind the circles according to whether current retrieval methods can achieve this or not. This will then provide impetus to improve retrieval methods for other conditions.

[Answer] Thank you for the thoughtful suggestions. To address the Reviewer's suggestion, we added detailed discussion of to what extent the X- and/or Ku-band retrieval algorithms has potential to achieve SWE based on recent literature in Section "5 Discussion and Conclusion".

"The OSSE results from this study suggest that the radar snow mission may be able to reduce the RMSE by 20% in the mountainous regions if the retrieval algorithm works in snowpack environments, up to 600 mm of deep SWE with 40% of TCF. This means that the algorithm developments must focus on enhancing the retrieval skill in both deep snowpack and moderate forest fractions. This could be achievable based on previous and ongoing efforts to demonstrate a sensitivity of X- and/or Ku-band signals to deep SWE in a forested environment. Recent studies found potentials of X-and/or Ku-band backscatters to estimate SWE by testing them in various snow environments. Borah et al. (2022) showed a sensitivity of co-polarization backscatters from airborne X- and Ku-band data to deep SWE more than 650 mm when using a bi-continuous dense media radiative transfer (DMRT) model. Santi et al. (2022) found that X-band backscattering coefficient from COSMO-SkyMed satellite can estimate deep SWE up to 800 mm using a retrieval approach based on machine learning methods. In preparation for ESA's

CoReH$_2$O mission, Montomoli et al. (2016) demonstrated that the X- and Ku-band SWE retrieval can provide in a forested region with the prior-knowledge of TCF and tree height if the TCF smaller than about 30% over a boreal forest site in Northern Finland. It is possible that the SWE retrieval could be useful in areas with denser TCFs if the detailed canopy structure, height, and forest types are available and adequate corrections can be applied (Tsang et al., 2022)."

*Borah, F. K., Tsang, L., Kang, D. K., Kim, E., Siqueira, P., Barros, A., & Durand, M.: Data Analysis and SWE Retrieval of Airborne SAR Data AT X Band and KU Bands, in 2022 IEEE International Geoscience and Remote Sensing Symposium, Kuala Lumpur, Malaysia, 17 - 22 July, 2022, 4252-4255, https://doi.org/10.1109/IGARSS46834.2022.9884965, 2022.*

*Montomoli, F., Macelloni, G., Brogioni, M., Lemmetyinen, J., Cohen, J., & Rott, H.: Observations and simulation of multifrequency SAR data over a snow-covered boreal forest. IEEE Journal of Selected Topics in Applied Earth Observations and Remote Sensing, 9(3), 1216-1228, https://doi.org/10.1109/JSTARS.2015.2417999, 2015.*

*Santi, E., De Gregorio, L., Pettinato, S., Cuozzo, G., Jacob, A., Notarnicola, C., Günther, D., Strasser, U., Cigna, F., Tapete, D. and Paloscia, S.: On the Use of COSMO-SkyMed X-Band SAR for Estimating Snow Water Equivalent in Alpine Areas: A Retrieval Approach Based on Machine Learning and Snow Models. IEEE Transactions on Geoscience and Remote Sensing, 60, 1-19. 10.1109/TGRS.2022.3191409, 2022.*

*Tsang, L., Durand, M., Derksen, C., Barros, A. P., Kang, D.-H., Lievens, H., Marshall, H.-P., Zhu, J., Johnson, J., King, J., Lemmetyinen, J., Sandells, M., Rutter, N., Siqueira, P., Nolin, A., Osmanoglu, B., Vuyovich, C., Kim, E. J., Taylor, D., Merkouriadi, I., Brucker, L., Navari, M., Dumont, M., Kelly, R., Kim, R. S., Liao, T.-H., and Xu, X.: Review Article: Global Monitoring of Snow Water Equivalent using High Frequency Radar Remote Sensing, The Cryosphere, 16, 3531–3573, https://doi.org/10.5194/tc-2021-295, 2021.*

Following on from this point, on line 100-101: 'to set priorities related algorithm developments.' Also line 361: 'thus different priorities for the algorithm development may be required by seasonal snow characteristics'. It would be really useful to have some concrete priorities identified from this study or at least a discussion of the factors. Fig 10 shows relative level of improvements but some may be out of reach so it's better to target the middle range improvements, or concentrate on montane forest because this is the largest % cover – at least for this site, but what about from a global perspective? The largest improvements are for tundra high forest fraction limits, but what does this mean in reality? How much of the tundra has dense forests according to the updated Sturm classification?

[Answer] Thank you for your valuable comment. Generally, priorities for the algorithm development are to improve capabilities to accurately retrieve SWE for both deep snow condition and forested regions which are long-standing challenges to snow remote sensing community.

We did calculate the percentage of areas for each snow class by VCF ranges over North America. The tundra areas with dense forests (> 40% TCF) are less than 1% of the entire tundra areas over North America, which is a similar proportion for this study domain. The boreal forest areas with less than 20%  and 40% TCF are 17.8 and 50.3% of entire boreal forest areas,

respectively, which are also pretty similar proportions for this domain (19.9 and 42.4%, respectively; Figure S3).

We acknowledge it is possible that our findings for a certain snow classification (such as Tundra) cannot be guaranteed to be applicable to other Tundra regions with different snow characteristics and landscape conditions (e.g., forest types). The best way to fully address this is to design a similar OSSE work for a larger study domain including multiple locations with the same classification but likely different snow and land characteristics. For this, we are currently working for a new OSSE study embracing the entire western U.S. and parts of north-central U.S.

We extended our previous discussion by including additional discussion from a global perspective in this study.

L445-450 "There are limitations to this study that may need to be considered in future research. First, the domain of this study (i.e., western Colorado) contains four seasonal snow classes and wide elevation ranges, enabling us to represent mountainous environments and quantify approximate performances in other regions that have similar snow regimes and land surface characteristics. However, we acknowledge that it is not enough to extrapolate our findings to global coverage of a future mission concept. For example, the snow condition of Tundra class in the domain of this study can be different from that of Tundra environments in Alaska. Further OSSE investigations with multiple domains in different snow climates, vegetation characteristics, and terrain complexity (e.g., steep vs. flat terrain) will complement current efforts."

Figure 3 seems to suggest there is more to be gained from improving the physics of the model than in developing assimilation techniques. Is this fair to say? For example, the OL in the mid elevation ranges overestimates melt by a lot in March. Why is the assimilation unable to recover this lost mass if the microwave data has SWE information? Do the OL and NR agree on the timing of the melt? If not, would you get better performance by detecting melt from the NR and forcing the DA model to be cold / use the observations when the NR is not melting but OL is.

[Answer] The reason why the assimilation was unable to recover the large melt in **Figure 3** is primarily the synthetic microwave data within the OSSE setting cannot provide accurate SWE information when snowpack is wet. In section 3.4 "Synthetic observation", we stated that "Because active microwave sensors cannot detect SWE if the snowpack contains liquid water (Matzler, 1987; Rott et al., 2010), synthetic observations are only assimilated when the snowpack does not include liquid water content (LWC). That is, when LWC values from the OL run are positive (> 0) at certain grids and periods, corresponding synthetic observations are not assimilated with the OL run.". Thus DA runs seemed to be unable to recover the large melt particularly during the melting season. It is hard to say that those related to limitations of the model physics or assimilation techniques.

Figure 4 – Is it fair to say there's no point in assimilating over bare ground? There seems to be negligible difference from the OL - why is this, especially as there are no SWE limits here?

[Answer] In this study domain, there are marginal areas over bare ground (TCF = 0%). Detailed areas of each seasonal snow class with TCF = 0% are provided in **Figure S1**. In most of those areas, very shallow snow accumulation typically occurred. Thus, there are negligible differences in the domain-averaged SWE between the OL and DAs with strict TCF limits.

[Figure]

**Figure S1**. Maps of each Sturm's seasonal snow class with different TCF thresholds (0, 10, 20, 40, 60, and 80%).

Line 297 / Fig 3. Why does the assimilation degrade the performance for shallow SWE?

[Answer] When creating the synthetic observations, an expected error of 30 mm of standard deviation (with zero mean) are applied to the SWE retrievals (Rott et al., 2010). Thus synthetic observations with those errors degraded the performance particularly for shallow SWE ranges.

We added some discussions as future directions.

L450-453 "Secondly, we applied a spatially constant error across the domain. While the error (30 mm of standard deviation with zero mean) was based on the expected uncertainty from previous studies (e.g., Rott et al., 2010), spatially and temporally dynamic error characteristics of the radar (e.g., multiplicative errors according to the amount of SWE) in OSSE experiments could improve the performance assessment."

In terms of specific comments for the paper, it would be good to address the following:

· Lines 40-45. Cut all these: certainly the acronyms (these aren't used again) but listing the satellites doesn't add anything to the paper so these can be removed. The fact that there are numerous instruments plus references are enough.

[Answer] We agreed with your suggestions. This part was removed.

· Remove TCF from the abstract and place the acronym definition at line 93

[Answer] Thank you for the comment. We removed TCF from the abstract and place this at the line.

· Line 93. Explain where the 40% comes from. Why is this considered achievable? It seems to be based on a LiDAR study, but this isn't referenced.

[Answer] We revisited the relevant references and decided that it would be more appropriate to modify this to 30% based on supporting references (e.g., Rott et al., 2012; Montomoli et al., 2016).

*Rott, H., Yueh, S. H., Cline, D. W., Duguay, C., Essery, R., Haas, C., Hélière, F., Kern, M., Macelloni, G., Malnes, E., Nagler, T., Pulliainen, J., Rebhan, H., and Thompson, A.: Cold regions hydrology high-resolution observatory for snow and cold land processes, Proc. IEEE, 98, 752–765, 2010.*

*Montomoli, F., Macelloni, G., Brogioni, M., Lemmetyinen, J., Cohen, J., & Rott, H.: Observations and simulation of multifrequency SAR data over a snow-covered boreal forest. IEEE Journal of Selected Topics in Applied Earth Observations and Remote Sensing, 9(3), 1216-1228, https://doi.org/10.1109/JSTARS.2015.2417999, 2015.*

· Line 100-101 - related -> related to

[Answer] Done.

· Line 119. Would be useful to state under what sensor assumptions here

[Answer] We added our sensor assumptions (X- and/or Ku-band at 1-km spatial resolution and 7-day repeat time) here.

L129-131 "The main objective of this study is to quantify the usefulness of X- and/or Ku-band volume-scattering SAR SWE retrievals at 1-km spatial resolution and approximately 7-day repeat time for improving spatially distributed characterization of snow conditions through an OSSE setup over a mountainous region of western Colorado."

· Figure 1. This needs a larger map to locate the region. Is this part of a larger modelling domain? If so, this should also be highlighted.

[Answer] Thank you for your good suggestion. We added a larger map to locate the study area.

[Figure]

· Line 145. Clarify NR and OL are at same spatial resolution

[Answer] We added a sentence to clarify the same resolution. "The NR and OL are designed at the same spatial resolution (1 km)."

· Lines 215-222. A supplementary figure demonstrating the flow of TAT-C derivation of the swaths would be helpful

[Answer] Thank you for your suggestion. We added a figure showing the masked swath using TAT-C in Supporting Information (Figure S3).

[Figure]

**Figure S3**. (a) an example of global daily TAT-C masked swath map and (b) examples of consequent five day TAT-C masked swath maps over the study domain

- Line 225. Why was the EnKF chosen for this work? (Is the EnKF acronym used after its definition? If not, it's not needed).

[Answer] We removed the EnKF acronym. Earlier studies have used the ensemble Kalman filter method when they did snowpack data assimilation work and OSSE developments (Kwon et al., 2021; Kumar et al., 2015; 2017). The ensemble Kalman filter method enables to flexibly characterize the model errors and to effectively handle non-linear dynamics and temporal discontinuities of observations (Lahmers et al., 2022). Thus we did use the ensemble Kalman filter and added an additional statement in the manuscript.

L241-243 "The ensemble Kalman filter method allows to flexibly characterize the model errors and to effectively handle non-linear dynamics and temporal discontinuities of observations (Kumar et al., 2015; Kwon et al., 2021; Lahmers et al., 2022; Cho et al., 2022)."

- *Kwon, Y., Yoon, Y., Forman, B. A., Kumar, S. V., & Wang, L.: Quantifying the observational requirements of a space-borne LiDAR snow mission. Journal of Hydrology, 601, 126709, 2021.*
- *Kumar, S. V., Peters-Lidard, C. D., Arsenault, K. R., Getirana, A., Mocko, D., & Liu, Y.: Quantifying the added value of snow cover area observations in passive microwave snow depth assimilation. Journal of Hydrometeorology, 16(4), 1736–1741. https://doi.org/10.1175/JHM-D-15-0021.1, 2015.*
- *Lahmers, T. M., Kumar, S. V., Rosen, D., Dugger, A., Gochis, D. J., Santanello, J. A., ... & Dunlap, R.: Assimilation of NASA's Airborne Snow Observatory Snow Measurements for Improved*

*Hydrological Modeling: A Case Study Enabled by the Coupled LIS/WRF-Hydro System. Water Resources Research, 58(3), e2021WR029867, 2022.*
· *Cho, E., Kwon, Y., Kumar, S. V., & Vuyovich, C. M.: Assimilation of airborne gamma observations provides utility for snow estimation in forested environments. Hydrology and Earth System Sciences Discussions, 1-26, https://doi.org/10.5194/hess-2022-332, 2022.*
·
· Equation 1 gives temporal RMSE – please could you include the equation for spatial RMSE in Fig 9 or adapt this one?

[Answer] Fig 9 is the spatial mean of "temporal" RMSEs. To clarify, we edited the caption of the Figure 9.

"**Figure 9** Spatial mean values of temporal RMSEs and the improved percentages (%) between the 24 DA experiments and the Nature run (NR) with different levels of deep snow and tree cover fraction (TCF) detection limits relative to the Open loop (OL) simulation"

· Fig 3 – would be nice to have box plots of SWE to see how much of the study area is affected by these limits. Also, please put this figure (+others) through a colour-blind checker. Perhaps make the OL a thicker line too.

[Answer] Thank you for your suggestions. I've checked this figure and others through a color-blind checker website (https://www.color-blindness.com/coblis-color-blindness-simulator/) and made sure if the lines are distinguishable each other (I attached a screenshot of a test). Also, I've made the OL a thicker line as below.

[Figure]

[Figure]

- Line 298-300. Please could you rephrase this? I think this is saying that the improvements come from better retrieval algorithms rather than the better spatial resolution with active, but I'm not sure.

[Answer] You understand correctly. I rephrased this to be clear.

This implies that even though a hypothetical mission can provide a finer spatial resolution SWE product (1 km) than existing passive microwave missions (e.g., 25 km), we could not achieve the better SWE estimates unless the hypothetical mission has better retrieval algorithms than that of passive microwave (e.g., 200 mm deep SWE limit and 20% of TCF).

- Line 306-307. Perhaps remove 'during a melting period' as the improvements are higher in the accumulation period (unless the intent is to highlight that any improvements go against expectations, in which case make more of it!)

[Answer] I agree with your suggestion. We removed "during a melting period".

- Figure 5. OL melting period spot needs to be black rather than grey.

[Answer] Thank you for your keen eye. We modified this.

[Figure]

- Lines 316-317. Use boxes to draw attention to the regions in Figure 6.

[Answer] Thank you. I added them.

[Figure]

· Fig 6. These appear have a different ratio to Fig 1. Please make sure there are the same number of x,y pixels in both. There also appears to be some striping artefacts in Fig 6f (thin white vertical strips of 1-2 pixels) – what is causing this?

[Answer] Thank you for pointing out this. The striping artefacts were caused in the process of converting statistic values to the raster type. We've updated the figure (attached above).

· Fig 7b – does white colour mean no change? Would be good to include this in the colourbar.

[Answer] The white color indicates areas with no DA occurrence. We've included the meaning of white color in the caption "white color indicates areas with no DA occurrence".

· Fig 8 – would be useful to extend the OL median up through (behind or dashed in front of) the other bars for easier comparison

[Answer] Good suggestion. We extended the OL median line up through the other bars. See the updated figure below.

[Figure]

· Line 397 – 'realistic orbital configurations for a volume scattering SAR mission developed using TAT-C': What is the (approximate?) repeat time in this study?

[Answer] 7-day repeat time; We added this.

· Line 422 – Please rephrase 'and there are values in the montane forest (17%) due to deep snow capability'

[Answer] I rephrased the sentence "there are some improvements in the montane forest (17%) due to the sensor capability for deep snowpack".

Finally, I'd like to commend the authors for Figs 2, 5 and 9 – these are informative / novel ways of representing the concepts and results (at least to me).

[Answer] We appreciate your compliment. Again, thank you for your time to give valuable comments on our manuscript.

---

## Author Comment (AC2)

**Reviewer #2: Anonymous Referee**

This study uses an OSSE to estimate improvement of SWE retrievals if satellite-based SAR data from an X- and/or Ku-band sensor was assimilated in an LSM. The authors use a calibrated version of the LSM (nature run) and compare it to an uncalibrated run (open loop) and a Data Assimilation run using synthetic observations with simulated extents using the TAT-C software. This paper is an excellent contribution to the overall field of SAR missions to retrieve SWE. It is very well structured and easy to read with excellent supporting figures. The fact that the analysis was done in a very different study area (Western Colorado), this study complements other similar studies (Garnaud et al., 2019). It provides information on what SWE limit and TCF the SAR mission should be able to detect for this specific domain in Colorado.

[Answer] Thank you for your excellent feedback and the valuable comments on our manuscript. We have carefully revised our manuscript based on your comments.

General comments:

One important limitation to this study that is not really discussed and I feel should be feasible with the current OSSE is the inherent geometrical limitations of SAR sensors (i.e. shadow/overlay) which complicates the retrieval of surface properties in mountain regions such as the region of interest in this study. The TAT-C software should allow to estimate the incidence angle and with the SRTM data, it should be feasible to mask out these blind spots. This evaluation might be outside the scope of this study, but I feel it should be discussed a bit further as a limitation of this study and be a future consideration. This would increase the number of masked grid cells that would not have SWE retrievals from satellite observations.

[Answer] Thank you for your valuable comments. We agree with the Reviewer's point. To address the concern, we added summarized discussions as the limitation in the manuscript as below.

L458-463 "Even though the OSSE of this study considers realistic sensor configurations for a volume-scattering SAR mission using the TAT-C software, there are inherent geometrical limitations of SAR sensors (i.e., shadow/overlay) which complicates the retrieval of surface properties in mountain regions such as the region of interest in this study. To design the OSSE more accurately, the geometrical observing gaps related to incidence angles of the SAR sensors and surface elevations should be accurately estimated. This may increase the number of masked grid cells that would not have SWE retrievals from hypothetical satellite observations."

To help reader understand the masked swath using TAT-C, we added a figure in Supporting Information (Figure S3).

[Figure]

To add to the other reviewer's comment: it would be useful to set the priorities of this study and give examples of what kind of mission would be relevant for these priorities since there is no "one-size fits all" mission. As mentioned the range of SWE values given for the Tundra class is not what you will find in other Tundra environments. Would a mission that would provide such improvement for the Tundra high SWE values work for other Tundra environments knowing the SWE values, snow stratigraphy (grain type/microstructure) and landscape conditions are very different? Adding some discussion on the specific snow conditions of the AOI would be relevant to this study.

[Answer] Thank you for making the reasonable point. We acknowledge it is possible that our findings for a certain snow classification (such as Tundra) cannot be guaranteed to be applicable to other Tundra regions with different snow stratigraphy and landscape conditions (e.g., forest types). The best way to fully address this is to design OSSE for a larger study domain including multiple locations with the same classification but likely different snow and land characteristics. For this, we are currently working for a new OSSE study embracing the entire western U.S. and parts of north-central U.S.

To better address the Reviewer's concern, we extended our existing discussion part by including additional discussion on specific snow conditions of the AOI in this study.

L445-450 "There are limitations to this study that may need to be considered in future research. First, the domain of this study (i.e., western Colorado) contains four seasonal snow classes and wide elevation ranges, enabling us to represent mountainous environments and quantify approximate performances in other regions that have similar snow regimes and land surface

characteristics. However, we acknowledge that it is not enough to extrapolate our findings to global coverage of a future mission concept. For example, the snow condition of Tundra class in the domain of this study can be different from that of Tundra environments in Alaska. Further OSSE investigations with multiple domains in different snow climates, vegetation characteristics, and terrain complexity (e.g., steep vs. flat terrain) will complement current efforts."

To add, this study only focuses on SAR retrieval from backscatter values. But what if the sensor has single-pass altimetry/interferometry capabilities? This would help to retrieve snow depths at least, especially during melt season from differential DEMs. Wouldn't that improve the estimation of SWE from the LSM? This might again be outside the scope of this study based on the priorities, but I feel this should be discussed as SAR missions are very rich data sources.

[Answer] Thank you for the Reviewer's comment. We are not sure what the comment "what if the sensor has single-pass altimetry/interferometry capabilities?" Does this mean either a hypothetical sensor has both SAR and altimetry/interferometry capabilities or altimetry/interferometry capabilities as one of the hypothetical sensor options to test within OSSE. Generally, we agree that the altimetry/interferometry sensors can help retrieve snow depths and have a potential to synergistically improve SWE estimates from LSMs using assimilation.

The altimetry/interferometry capabilities (such as ICESat-2) for snow depth retrievals within an OSSE framework have been examined in another recent study (Kwon et al., 2021). They found that the smaller number of available snow depth observations given the narrow ICESat-2 sampling geometry led to relatively small improvement of SWE estimates.

Specific Comments:

L.39-44: No need to list the different PMW sensors here, I would keep "Historically, a series of satellite-based passive microwave radiometers have been used to develop spatially distributed snow depth and SWE information (Cho et al., 2017; Derksen et al., 2005; Foster et al., 2005; Vuyovich et al., 2014).

[Answer] Agreed. We have removed the list of the PMW sensors.

l.220: to make this OSSE more realistic, what would be the incidence angle range of such a SAR mission configuration?

[Answer] The reviewer makes a reasonable point. We agreed that the result would be more realistic if we consider the incidence angle ranges in the OSSE framework.

Fig 4.: provide the TCF ranges for the different elevations. I suspect there is not much TCF over low and mid elevations where there is not much improvements in runs with more TCF capability.

[Answer] Thank you for the suggestion. We added the TCF percentages for the different elevations in Supporting Information (Table S2). As you expected, for low elevations, low TCF

areas were dominant (e.g., 86% of areas with TCF up to 20%), resulting in small improvements with more TCF capability. For mid elevations, there were still small improvements, even though more than half of the areas are with high TCFs (e.g., 51% areas with TCFs above 20%)

We mentioned the inclusion in the caption of Figure 4 like below.

"Figure 4. Domain-average SWE comparison between NR, OL, and DA experiments with different levels of detection capability in areas with bare ground and tree cover fraction (TCF) limits up to 10, 20, 40, 60, and 80%. The areal proportion of TCFs for three elevation ranges are provided in Table S2."

**Table S2.** The areal proportion of the different TCF ranges for three elevation ranges

|  | Low elev. (0-2500 m) | Mid elev. (2500-3000 m) | High elev. (3000-4000 m) |
|---|---|---|---|
| TCF = 0% | 14 | 3.5 | 0.6 |
| TCF up to 10% | 76 | 36 | 25 |
| TCF up to 20% | 86 | 49 | 39 |
| TCF up to 40% | 96 | 73 | 69 |
| TCF up to 60% | 99 | 93 | 93 |
| TCF up to 80% | 100 | 100 | 100 |

Fig 7.: Change RMSD to RMSE

[Answer] We changed this. Thank you.

---

## Author Response (AR2)

Reviewer #1

Dear Dr. Cho and Colleagues,

Thank you for your revisions to this manuscript. This forms a worthwhile contribution to the design of future satellite missions and recommendations on where to focus research efforts. This is suitable for publication subject to the following minor corrections:

Answer: Thank you, Dr. Mel Sandells, for your thoughtful feedback on the manuscript. We have revised our manuscript based on your corrections.

Line 290: 'Averaged for given areas' - is this domain averaged as in line 46 or something else - please clarify.

Answer: Thank you for pointing out. The statement was removed. We believe that the point we want to make here was explained well in the previous sentence "Figure 5 provides a comprehensive comparison of RMSEs and percent improvement calculated by the time series of domain-averaged SWE between all DA scenarios and the OL simulation relative to the NR.".

Figure 6: Boxes added in authors' response has not been included in the paper - please replace with the new figure.

Answer: We replace the old figure with the new one in the manuscript. Thank you.

Figure 7: RMSD -> RMSE difference.

Answer: Edited.